# The Modeling and Forecasting of Carabid Beetle Distribution in Northwestern China

**DOI:** 10.3390/insects12020168

**Published:** 2021-02-16

**Authors:** Xueqin Liu, Hui Wang, Dahan He, Xinpu Wang, Ming Bai

**Affiliations:** 1School of Agriculture, Ningxia University, Yinchuan 750021, China; liuxueqin@nxu.edu.cn (X.L.); hedahan@163.com (D.H.); 2Institute of Green Manure, Yan’an Academy of Agricultural Sciences, Yan’an 716000, China; wangbin1986221@126.com; 3Key Laboratory of Zoological Systematics and Evolution, Institute of Zoology, Chinese Academy of Sciences, Beijing 100101, China

**Keywords:** conservation, indicators, species distribution modeling, species richness, generalized additive models, steppe, China

## Abstract

**Simple Summary:**

The relationship between species and environment are an important basis for the study of biodiversity. Most researchers have found the distribution of indicator insects such as carabid beetle at the local community scale; however, a few studies on the distribution of indicator insects in grassland in China. Here, we used Generalized Additive Models (GAM) to predict temperate steppe of northwestern China carabid beetle species richness distribution, and to determine the possible underlying causal factors. Predicted values of beetle richness ranged from 3 to 12. The diversity hotspots are located in the southwest, south and southeast of the study area which have moist environment, the carabid beetle is mainly influenced by temperature and precipitation. The results underline the importance of management and conservation strategies for grassland and also provides evidence for assessing beetle diversity in temperature steppe.

**Abstract:**

Beetles are key insect species in global biodiversity and play a significant role in steppe ecosystems. In the temperate steppe of China, the increasing degeneration of the grasslands threatens beetle species and their habitat. Using Generalized Additive Models (GAMs), we aimed to predict and map beetle richness patterns within the temperate steppe of Ningxia (China). We tested 19 environmental predictors including climate, topography, soil moisture and space as well as vegetation. Climatic variables (temperature, precipitation, soil temperature) consistently appeared among the most important predictors for beetle groups modeled. GAM generated predictive cartography for the study area. Our models explained a significant percentage of the variation in carabid beetle richness (79.8%), carabid beetle richness distribution seems to be mainly influenced by temperature and precipitation. The results have important implications for management and conservation strategies and also provides evidence for assessing and making predictions of beetle diversity across the steppe.

## 1. Introduction

The relationship between species and environment has always been a central topic in ecological research, the spatial distribution of species is closely related to environment [1]. Climate and human activity as a main threat to global biodiversity is increasing, the Global Assessment Report on Biodiversity and Ecosystem Services of IPBES [2] recorded that one million animal and plant species are facing extinction at present, destroying ecosystem functions and services. Therefore, efficient management tools are urgently needed to protect biodiversity and maintain global ecosystem functioning and services [3,4,5].

One management tool is to improve the predictive ability of biodiversity distribution models, such as niche models, species distribution and habitat suitability [6,7]. Studies have shown that a diverse ecosystem characteristic have been predicted from the main environmental drivers, including species distribution, richness, biodiversity value and soil characteristics and also proved models was very valuable to biodiversity research and mostly useful for policy makers [8,9]. Ideally, planning biodiversity conservation should integrate as many taxonomic groups as possible, including indicator insect groups such as the carabid beetles. Carabid beetles are an essential component of global biodiversity and play a vital role in global ecosystems (e.g., indicator, predators) [10,11]. Carabids are commonly used to studies grassland management, as their ecology is well-known [12,13]. Beetles are sensitive to environmental change and perceived good for agriculture, so ecologists and taxonomists have turned to carabid beetles to test ecological research questions, thus beetle currently faced numerous threats [14,15]. The main threats come from the land use and grassland degradation, which leads to the loss and degradation of beetle habitats [16,17,18], inducing changes in beetle community composition [19,20,21]. The relationships between the environment and beetle communities are complex phenomena. Generalized Additive Models (GAM) can best describe these linear or nonlinear relationships between beetle and environment by using nonparametric smoothing terms [22]. Hence, based on the advantage of GAM for accommodating nonlinear relationships between variables, GAM is expected to efficiently model the relationship between environmental variables and beetle diversity and provide reliable results.

Specifically, in China, grassland vegetation, which accounts for 80% of the steppe, is degrading rapidly due to climate change and human activities that are changing productive steppe into barren land and desert [23]. The degradation will have a huge impact on steppe biodiversity and there is an urgent need to study the ecology of groups that may serve as an indicator insect. Modelling procedures can be useful tools to provide robust and accurate estimates of current and future distributions, abundance, and the population dynamics of species, and these can be directly applied to conservation and management practices [24].

Carabid beetles (Coleoptera: Carabidae) represent an abundant and diverse insect group [25,26] and account for an important fraction of total diversity [27,28,29]. The vital contributions of this taxonomic group to ecosystem management have been largely recorded, for example, they are used as an index of habitat restoration, land use, degree of urbanization and an indicator of shrub erosion in the steppe [13,30,31]. In addition, these species can prey on a large number of pest insects [32]. Despite beetles having proven ecological benefits and service in grassland ecosystems, their distribution patterns have been poorly recorded [33,34] which currently is endangering the maintenance of their ecological roles. Therefore, it is essential to integrate information about beetle diversity and distribution into grassland sustainable development strategies to improve or at least conserve their biodiversity and ecosystem services in steppe regions.

However, field investigations are challenging because of remoteness, inaccessibility of many steppe areas and shortage of staff. Species Distribution models (SDMs) provide a cost-effective tool to overcome these limitations and remotely assess biodiversity over large areas at regular intervals over time [35,36]. SDMs have been widely used to assess distribution and diversity patterns of different organisms [37,38,39]. Increasing numbers of studies use SDMs to assess, model, predict or map species’ distribution and analyze biodiversity [40,41,42].

Here, we used Generalized Additive Models (GAM) to predict and map beetle richness patterns [43]. In this study, we contribute to model the species richness in unmeasured area to promote grassland management and develop a conservation policy strategy for governments and also to determine the main driving factor of beetle’s distribution. Our overall aim is to the conserve beetle biodiversity and maintain their ecosystem services in the grassland regions which form the main ecology in northwestern China.

## 2. Materials and Methods

### 2.1. Study Area

This study was undertaken in two regions of Ningxia Hui Autonomous Region which represent a temperate steppe ecosystem in northwestern China and comprised between 36° north (N) and 38° N and between 105° east (E) and 108° E.
(1)Yanchi region, characterized by a cold, semi-arid continental monsoon-influenced climate, with a mean annual temperature of 5.7 °C and mean annual precipitation of 200 mm [44]. The soil was of sierozem and the representative vegetation is *Agropyron mongolicum*, *Artemisia desertorum*, *Lespedez adavurica* and *Artemisia blepharolepis*.(2)Guanyuan region, characterized by a semi-arid continental monsoon-influenced climate, with a mean annual temperature of 7 °C and mean annual precipitation of 400 mm [44]. The soil was of black thorn and brown and the representative vegetation is *Stipa bungeana*, *Artemisia frigida*, *Potentilla acaulis* and *Stipa grandis*.

### 2.2. Beetle Data

The beetle data used in this study was from the steppe of northwestern of China which was sampled in 2017, 2018 and 2019; we selected 124 sampling sites and at each sampling site placed at random five pitfall traps (separated by at least five meters from each other), all sampling site were separated by at least 150 m in order to avoid possible autocorrelation. Samples were taken from May to September every year, which allowed us to obtain a good representation of carabid richness. We accounted for number of beetle once a month and take the average of five times for analysis. We divided each study area into 10 × 10 km^2^ grid squares in order to discriminate adequately-surveyed grid squares, the value of each 100 km^2^ grid, well surveyed were identified and recorded of all species observed (Figure 1). Five pitfall traps (400 mL capacity, 7.5 cm diameter, filled with 40–60 mL of a 2:1:1:20 vinegar, sugar, alcohol and water solution and covered with a suspended opaque plastic roof) were placed at each site and collected three days later. Trapped beetles were stored in 75% ethanol and transported to the laboratory for identification to species level with the aid of a taxonomist expert in carabid beetles (Prof. H. Liang, see Acknowledgments). Analyses were conducted using the pooled data from the average values every year.

### 2.3. Environmental, Spatial and Climatic Data

At each sampling site, we selected a 1 × 1 m quadrat frame (the habitat around each site is very homogeneous) and measured plant dry biomass (PB), cover (PC, %), density (PD), height (PHe), plant richness (PSD), litter dry mass (SL, g/m^2^), soil moisture (SM, %), bulk density (SBD), and soil temperature (ST), soil organic matter (C), total phosphorus (P), total nitrogen (N), pH value (pH). Above of vegetation and soil were measured once a month. The soil moisture and temperature (underground 10 cm) were measured by a portable soil water potential temperature tester (TRS-II, China). The climate data including the maximum and minimum monthly mean temperature (T, t), the annual mean temperature (TM), and annual precipitation (p) were extracted climatic dataset (www.worldclim.org, accessed on 1 December 2020). The spatial data (longitude (Lon) and latitude (Lat)) and the geographical data (altitude (Alt)) were measured by GPS (G128BD, China). The information of variables saw Table A1.

### 2.4. Data Processing and Statistical Analyses

Species activity density was calculated as the number of individuals per square meter; Margalef index was calculated using the formula: $\left(S-1\right)∕ln^{N}$, S is the number of species, where *n* is the number of collected individuals per square, species richness was expressed as the number of beetle species in a given grid cell. All analyses were performed in R v.4.0.3.

### 2.5. Model Building

A total of 19 potential predictors were preselected. First, the method of variance inflation factor was used to select the most important environmental predictors for each response variable and the largest variable was deleted in turn, that is, the collinear environmental factors were deleted, until all variables were less than 10 from the ‘car’ package in R (v.4.0.3). This package (vif function) can help us to identify and keep important relevant predictors in our models. Second, the predictor factors were further refined by using the Pearson correlation coefficient to identify highly correlated variables (|r|) > 0.7) and avoid the inclusion of redundant variables in our models. The goodness-factor for the competing functions was measured by an F-ratio with a 5% significance level and the non-significant factors were removed (Table A2). Third, a backward stepwise procedure was used to enter the variables into the model [45]. The step model was used to detect the lowest AIC value, and the optimal environmental factor was automatically selected. When “none” is at the top, it means the end of model selection. The number of beetle richness as dependent variable in order to remove the non-significant spatial terms. The significant spatial terms (*p* < 0.05) were retained. 

The sampling is stratified random sampling. The soil and environmental factors of the unsampled grids were interpolated based on the statistical relationship among the surrounding measured site in each year. To compare different result for GAMs and GLM, an independent dataset was used. The data from 2017 to 2018 for training set and data from 2019 as test set were randomly chosen to evaluate offset between predicted values of the model and the original values. For model validation, we used a correlation coefficient between predictive and real species richness values. The higher the value of the correlation coefficient, the higher the predictive power of the model.

### 2.6. Model Fitting and Selection

Species distributions were modelled with generalized linear model (GLM) and Generalized Additive Models (GAM) in order to seek the best model. GLMs are defined by the response distribution and a link function. The structure is as follows:(1)$$g\left(\mu_{i}\right)=x_{i}{}^{T}\beta$$
where g is the differentiable and monotonic link function, $\mu_{i}=E\left(Y_{i}\right)$, $x_{i}$ is the explanatory variable for the ith response variable, β is a vector of the parameters. The log-transformation has been found to be fit for many situations and data sources, despite its great generality, the GLM has serious limitations. Generally, AIC is usually used criterion for model selection when GLMs/GAMs are used to estimate species richness [46].

GAM is an extension of the Generalized Linear Modelling (GLM; [47]) which uses a link function to establish a relationship between the mean of the response variable and a ‘smoothed’ function of the explanatory variable [48,49]. GAMs can model highly non-linear and non-monotonic relationships between the response and the set of explanatory variables. GAM has been widely applied in ecological research, as shown by the growing number of published papers incorporating modern regression [50,51,52,53]. GAM implemented in the mgcv package in R (4.0.3). The most optimum model was selected with the lowest Akaike Information Criterion (AIC) and residual deviance [54]. The general form is as follows:(2)$$g\left(\mu \left(Y\right)\right)=\beta_{0}+f_{1}\left(x_{1}\right)+\cdots +f_{m}\left(x_{m}\right)$$
where g(.) is the connection function; μ(Y) is the expected value of the response variable Y; $\beta_{0}$ is a constant; and $f_{m}\left(.\right)$ is a smooth function of the explanatory variable $x_{m}$.

Poisson distribution for species richness was used. To avoid data overfitting, the basic dimension was defined as k = 4. In order to improve model performance, the values of the parameters of the GAM algorithm were optimized independently for each model, selecting those that minimized the AIC; this step has been considered as providing an estimate of model reliability. For model assessment, the evidence ratio, AIC and minimized generalized cross validation (GCV) score were applied. The smaller the values of GCV, the better the models fit [55].

## 3. Results

### 3.1. Population Size

The mean number of the carabid beetle (number of beetles in each grid cell) was 39.88 ± 79.4 individuals/km^2^, the mean number of beetle species was 8.92 ± 1.11 individuals/km^2^. On the other hand, species activity density was 0.897 individuals/m^2^ and the Margalef index was 8.54.

### 3.2. Fitted Model

Compare to GLM, results for comparing performances are shown in Table 1. According the AIC criterion results showed that GAMs has a lower score compared to GLM (AIC-GLM = 609.54, AIC-GAM = 598.04). In addition, R^2^ (0.774), *p*-value (<0.001) and correlation coefficient (0.923) also indicated that GAM has a high quality for model performance, GAMs fitted the observed data as much as possible by enabling the smooth effects of the continuous predictors as well as the spatial structure of the data (Table 1).

The stepwise algorithm parameters used to develop the models, as well as F value and P estimates are listed in Table 1. Seven environmental variables: maximum mean temperature, mean annual precipitation, latitude, longitude, plant density, soil bulk density, soil temperature, and PH value were statistically significant after the collinearity analysis of all environmental factors (Table A3). Among the seven environmental variables, the maximum mean temperature (F = 5.336, *p* < 0.00046), mean annual precipitation (F = 9.031, *p* < 0.05) and soil temperature (F = 5.336, *p* < 0.001) had statistically significant effects on species, whereas soil and climatic variables consistently appeared among the most important predictors for the richness of beetle groups modeled (Table 2). General trends seem to be very well identified by the GAM. The GAM parameters used to develop the models, as well as adjust the fit factor (R2), generalized cross validation (GCV) and deviance explained are listed in Table 2. By comparing the different explanation variable of function of GAM results, selection of model variance explained the largest volume, minimum generalized cross validation, F test (*p*) model of the highest accuracy rate value as the optimal model, in general, when the rate of F test value (*p* < 0.05), indicating that explain the response variables affect significantly, if adjust the fitting coefficient (R^2^) is greater than 0.5, that model has good stability and effectively explains the response variables and explains the relationship between the variables. Among the seven predictor variables, the full model was the best adjusted explaining 79.8% of the variation, the GCV was 0.062 and the adjust the fitting coefficient (R^2^) was 0.774, our model showed a good predictive performance for beetle richness and the best model was log (species richness) = s(T) + s(*p*) + s(ST). Plots of the relationship of predicted richness distribution and environment variables are shown in Figure 2; the distribution of beetle richness mainly depends on the maximum mean temperature, mean annual precipitation and soil temperature. The relationship between maximum mean temperatures, mean annual precipitation and spatial distribution of beetle were complex, but it was positively correlated with soil temperature change (Figure 2).

### 3.3. Predictive Mapping

The results from the predictive mapping of the beetle richness at the spatial level are shown in Figure 3. Predicted values of beetle richness ranged from 3 to 12. Three diversity hotspots are located in the southwest, south and southeast of the study area. The statistics of the coefficient of variation showed that, overall, predictions from individual GAMs of beetle richness at the spatial level were stable.

## 4. Discussion

SDMs are important ecological tools for conservation planning and management [56]. The present study demonstrates the efficacy of SDMs to assess the species richness of beetle in grassland. GAM analysis suggested that the three most important factors, which showed the largest effect of the beetle richness, were mean maximum temperature, annual precipitation and soil temperature. The soil temperature changed with the temperature, the result supported the factors determining beetle life cycles include variation in temperature and rainfall [57]. Our model explained a significant fraction (0.77) of the variation in beetle richness. Our study also provides a potential methodology for conservation of the species groups.

The model of distribution of beetle richness helps understanding the relationships between beetles and their environment, and thus is useful for protection and management purposes. GAM used smooth functions to deal with nonlinear relationships between the response variable and explanatory variables, increasing evidence that GAM is likely to be more suitable to estimate the distribution of species richness [43,58]. GAMs showed good performance for species richness estimation in the present study (79.8%), and robustly explain the relationship between the variables and species richness. Our study demonstrated that climatic factors, especially temperature and precipitation are the important environmental factors generating richness patterns of the beetle group. Both temperature and precipitation show a curvilinear relationship with species richness and had a significant effect on beetle species, this result agrees with those obtained in earlier studies of carabid beetles [19]. Although, the suitable maximum annual temperature value is around 20 °C, the effect may be reflecting the effect of temperatures of the warmest months which can limit the activity of beetles according to their tolerance to desiccation. Because most carabids are active on the ground, their body temperature depends directly on the ambient temperature and it is known that species activity can be stimulated by temperature [59]. The importance of precipitation can be explained by the free ranging life style of immature larval stages. The amount of precipitation enhances the aboveground vegetation biomass [60], and vegetation provides food and shelter (from the environment and predators) for the herbivorous species. However, precipitation hinders the survival of some of these species when it exceeds the threshold value. The soil temperature has a significant correlation with carabid beetle richness. Because some species lay eggs in burrows, and others overwinter as larvae or as adults in the soil, the soil temperature can stimulate or hinder species activity. We suggest that lower observed beetle richness may be due to the higher temperature, precipitation, and correlated soil temperature in those areas.

Carabid beetles also respond to microhabitat conditions. Carabids perceived microhabitat variation and selected niches accordingly [61], increasing evidence that management of microhabitats is a key tool for conserving ecosystem function. The Carabid beetle fauna in the steppe ecosystem in Ningxia Hui Autonomous Region has been recorded over ten years, but the beetle information only shows where the entomologist sampled and the composition of beetle [62,63]. In this study, we have shown that with a reasonable sampling distribution, predictive variables for species richness can be derived efficiently from GIS-based data for areas in which species inventories have not yet been conducted, and a reliable forecasted map of species richness may be obtained. The forecasted map can be used to plan and carry out new, targeted studies and regional surveys thus saving on the resources needed for large-scale surveys. It is very expensive and may be impractical to sample all poorly surveyed areas. The forecasted map also can provide an opportunity to manage these habitats and conserved carabid taxa. 

Carabid beetles live in moist habitats and are excellent model species on research of ecological and conservation theory [64,65,66]. The 3rd International Carabidologists’ Meeting emphasized that it is needed to concern on the effects of habitat loss and fragmentation on dynamics of beetle population if wise decisions are to be made regarding conservation and land-use [67]. These beetles readily respond to disturbances and management. Our results show that it is quick and inexpensive to employ forecasting models using simple environmental variables and adequately sampled areas to produce an estimate of the spatial distribution of species richness and obtain reasonable biogeographic patterns. Relating biological data to environmental variables without adding geographic position as a model predictor sometimes overestimates the actual species richness [68,69]. Our results demonstrate that elaborating predictive models using simple environmental variables is quicker and less expensive when based on the concept of adequately sampled areas. Consequently, our model for the species richness of carabid beetle distribution provides a good substitute for information that could not be provided otherwise in the coming years. This information will focus sampling efforts, and also inform management and conservation strategies.

## 5. Conclusions

Our models explained a significant fraction of the variation in beetle richness (79.8%), and predictive mapping of carabid beetle richness at the spatial level helped us to identify important variables determining the richness of beetle species. For the carabid group of beetles, species richness variation was influenced primarily by the climatic factors of maximum temperature and precipitation. If the survival of carabid species is constrained by temperature and precipitation (few species can tolerate high temperature and precipitation), we argue that species richness variation in the steppe of northwestern China is due mainly to the failure of many species to go beyond determined temperature and precipitation range limits. Thus, the regions richest in species are those with a temperature and precipitation compatible with the maintenance of populations.

## Figures and Tables

**Figure 1 insects-12-00168-f001:**
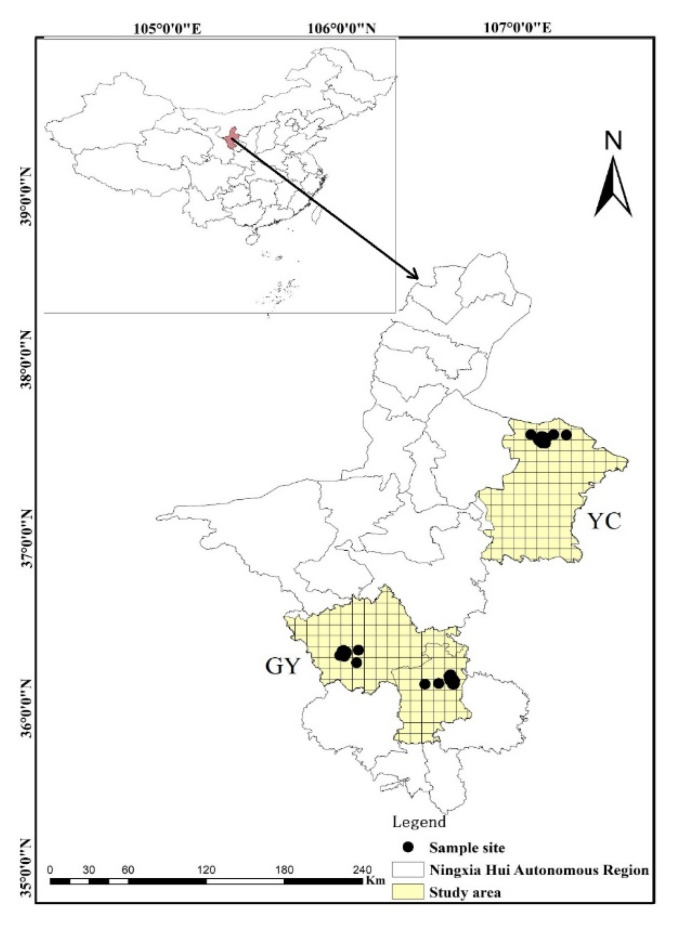
Study areas and sampling site (n = 124) in the Ningxia Hui Autonomous Region, northwestern China (YC, Yanchi region; GY, Guyuan region).

**Figure 2 insects-12-00168-f002:**
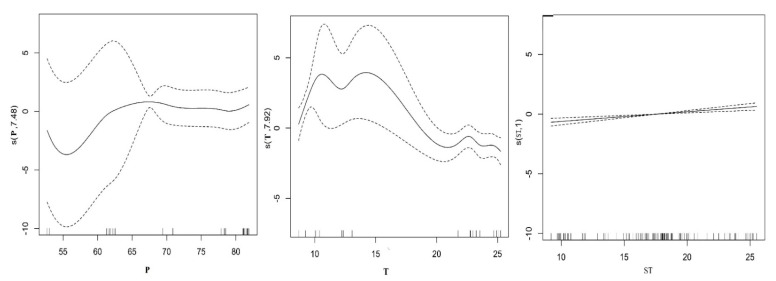
Response curves of carabid beetle and factors kept in the GAM analysis. The vertical axes are expressed in logits, and the value (s) represents the smoothing fitting value of explanatory variable of beetle species, the ordinate in parentheses is the estimated degrees of freedom. Solid curves are the function estimates, and dashed curves delimit the 95% confidence intervals for each function.

**Figure 3 insects-12-00168-f003:**
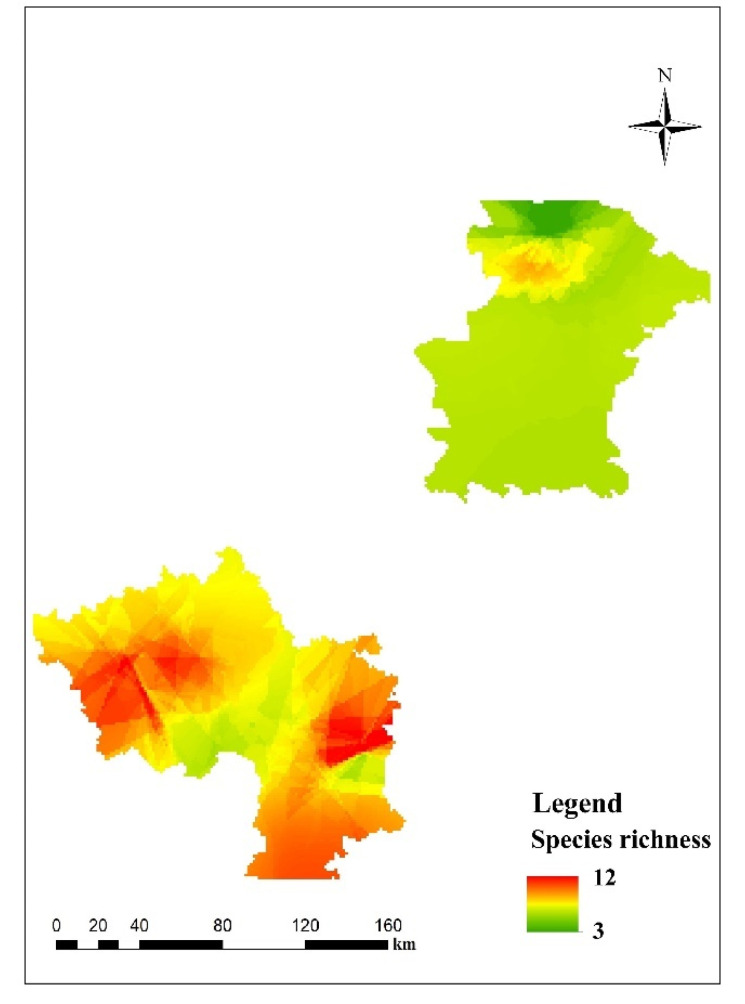
Predictive beetle richness for the study area. Red colors correspond to areas predicted to be species rich, while yellow and green colors correspond to areas predicted to harbor intermediate and low levels of richness.

**Table 1 insects-12-00168-t001:** Summary of stepwise of the GAM and GLM of variables.

Model	GLM				GAM		
Variables	Deviance Residual	Df. Residual Deviance	F	*p*	df	F	*p*
Latitude	11.53	148.67	5.872	0.635	1	1.436	0.23360
Maximum mean temperature	3.35	144.17	6.725	0.005 ***	3.734	5.336	0.00060 ***
Mean annual precipitation	11.2	137.47	0.094	0.925	6.071	9.013	<0.05 *
Plant density	9.67	138.05	−2.406	0.016 *	5.861	0.773	0.49420
Soil bulk density	6.45	117.9	0.465	0.465	6.428	4.333	0.03988
Soil temperature	29.42	137.87	1.967	0.049 *	1	3.285	0.00499 ***
PH value	18.43	106.27	5.660	<0.001 ***	1	1.486	0.22563
AIC	609.54				598.04		
R^2^	0.682				0.774		
*p*-value	<0.001				<0.001		
correlation coefficient	0.907				0.923		

*** *p* < 0.001; * *p* < 0.05.

**Table 2 insects-12-00168-t002:** The result of GAM of variables to build the models for the beetle richness. Variable codes as in Table A1.

Environmental Factor	df	F	Adjust the Fit Factor (R^2^)	Generalized Cross Validation (GCV)	Deviance Explained (%)
p	8.991	<0.001	0.727	0.073	74.7%
T	8.999	<0.001	0.694	0.08	71.6%
ST	8.626	<0.001	0.455	0.1449	49%
p+T	8.856	<0.001	0.729	0.0737	75.2%
p+ST	8.991	<0.001	0.753	0.066	77.3%
T+ST	8.981	<0.001	0.758	0.068	78.5%
p+T+ST	8.689	<0.001	0.774	0.062	79.8%

## Data Availability

The data presented in this study are available on request from the corresponding author. The data are not publicly available due to uncompleted subject.

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
