# Peer review of "The Modeling and Forecasting of Carabid Beetle Distribution in Northwestern China"

_insects, 2021, doi:10.3390/insects12020168_

Round 1
Reviewer 1 Report
I reviewed the manuscript entitled The modeling and forecasting of Carabid beetle distribution in 2 northwestern China”. Authors perform a model of potential distribution of species richness of carabid beetle communities in the steppes from northwestern China, related by environmental predictors. The manuscript includes a considerable effort for modeling, but I consider that the manuscript must be improved taking into account the following considerations:
.- Major considerations (Mc):
Mc1.- About predictors dataset.- Authors use a dataset of environmental predictors composed by two kind of variables: vegetation and soil variables, climatic variables and geo-topographical variables. The first problem of it is the no definition of a homogenous work scale: 1 m2 for the first, and unknown for the other two. About climatic variables, authors do not indicate the time period of meteorological records use for calculate the variables, neither the distance of meteorological stations from work area. No indication there is about if climatic variables were calculated by Authors or were obtained by a provider public or private. Neither work scale of spatial model are referred.
I am not sure how Authors calculated the prediction map of Figure 3, because no explanation about of the predictor data set are included. Plus, I can not understand where it they get the extrapolation of soil temperature for the entire work area, as well as, the other non-climatic variables.
Mc 2.- About modeling procedure. I’m in agreement that GAM is a good and flexible method for obtain distribution models, but always is recommendable the use of at least other modeling methodology for contrasting, as EMFA, GLM or MAXENT, all available in R. The comparison between different spatial results from different models usually helps to understand a more actual potential distribution.
Mc 3.- Discussion and conclusions need a deeply revision, because include results, as the lines 231 to 236, or 268 to 270. Conclusions is an extension of Discussion and do not contain true conclusions.
.- Minor considerations (mc):
mc 1.- Simple summary are not necessary, because repeat the Abstract contents. This sections are not consider in the Instructions for authors of Insects journal. I think that these section should will be removed.
mc 2.- Figure 1 need a strong improvement for be adapted to the minimum standard quality of a geographic information. I think that they should include a basis cartography, information of geographic scale, geographic reference, cardinal reference, general location in a general map of eastern Asia, etc.
mc 3.- Beetle data. Authors wrote that samples were taken from May to September every year (2017-2019), in sampling of 3 days in 124 plots, but I do not understand how often times repeat this sampling between May and September, once, twice? Monthly, weekly? I think that these data is relevant to understand the quality of the insect data set, composed by 26 species, a not too high number of species considering the apparent big work area.
mc 4.- I miss a table with information of predictors variables: units, maximum and minimum values, scale, etc…
mc 5.- Results. I think that the first results that should be shown are the species richness data for area (mean +- sd). Why authors do not use other more informative biodiversity index as Shannon Index?
mc 6.- Figure 2. What mean “The vertical axes, expressed in logits, indicate the relative influence of each explanatory variable on the prediction on the base of partial residuals” ? These graphs are not too understandable with negative values in the Y axes.
mc 7.- Accordingly with the instruction of the Insect journal, units should be referred in SI (km), not imperial (miles).
I think that the paper is not suitable for publication in the present stage, but I also think that it could be improved.
Also I encourage to the authors to beside use the useful and free climatic dataset WorldClim (https://www.worldclim.org/), whose could improve their model. Also I encourage to try of include other models as ENFA, GLM or MAXENT.
Author Response
Dear review:
Please see the attachment, thank you.

Reviewer 2 Report
General comments
In this paper the authors aimed to predict and map beetle richness patterns within the temperate steppe of Ningxia (China) by using Generalized Additive Models (GAM). The paper is well organized, has clear objectives and the drawn conclusions are coherent with the obtained results.
Their results have important implications for management and conservation strategies and also provides evidence for assessing and making predictions of beetle diversity across the steppe.
Specific comments
Line 22: It should be…Using Generalized Additive Models (GAMs)
Lines 25 – 26: To delete
Lines 33: Please, add other key words..for example: Generalized Additive Models, steppe, China
Lines 43 – 44: I think that you should add some recent references to support this your sentence “One management tool is to improve the predictive ability of biodiversity distribution models, such as niche models, species distribution and habitat suitability.” I would like to suggest:
Raffini, F., et al. (2020). From Nucleotides to Satellite Imagery: Approaches to Identify and Manage the Invasive Pathogen Xylella fastidiosa and Its Insect Vectors in Europe. Sustainability, 12(11), 4508.
Vila-Viçosa, et al. (2020). Combining Satellite Remote Sensing and Climate Data in Species Distribution Models to Improve the Conservation of Iberian White Oaks (Quercus L.). ISPRS International Journal of Geo-Information, 9(12), 735.
Lines 57 – 63: To describe why this method should be better for your study
Line 79: It should be Species Distribution Models (SDMs)
Lines 43 – 44: I think that you should add some recent references to support this your sentence “SDMs have been widely used to assess distribution and diversity patterns of different organisms [35]” I would like to suggest:
Bertolino, S., et al. 2020. Spatially-explicit models as tools for implementing effective management strategies for invasive alien mammals. Mammal Review, 50: 187-199. https://doi.org/10.1111/mam.12185
Cerrejón, C., et al. (2020). Predictive mapping of bryophyte richness patterns in boreal forests using species distribution models and remote sensing data. Ecological Indicators, 119, 106826.
Lines 93 – 34: Please, add the geographical coordinates
Line 154: To move this table in the supplementary materials
Line 223: SDMs instead of Species distribution models
Author Response

(The authors gave the same response as above.)

Round 2
Reviewer 1 Report
I reviewed the second version of the manuscript entitled The modeling and forecasting of Carabid beetle distribution in 2 northwestern China”. In my first revision I considered that manuscript was not suitable for publication and I proposed several recommendations to the authors for improve it. Reading the new version of the manuscript, I find scarce consideration of recommended improvements.
.- Major recommended considerations (Mc):
Mc1.- About predictors dataset.- Authors use a dataset of environmental predictors composed by two kind of variables: vegetation and soil variables, climatic variables and geo-topographical variables. The first problem of it is the no definition of a homogenous work scale: 1 m2 for the first, and unknown for the other two. About climatic variables, authors do not indicate the time period of meteorological records use for calculate the variables, neither the distance of meteorological stations from work area. No indication there is about if climatic variables were calculated by Authors or were obtained by a provider public or private. Neither work scale of spatial model are referred.
Authors do not amenably improved this section. They do not explain how interpolate soil measures in sampled from 124 samples taken in a detailed scale of 1x1 m, to a study area of tens of thousands square km. They neither explain the period of meterorological records for the calculation climate data. Author indicate that local meteorological stations are located at 6 km far of the sample site. But, there are 124 sample site, so this is very imprecise. No indication about meteorological data provider is included, neither how the climate model is builded for interpolate to the entire study area.
I am continue no understand how Authors calculated the prediction map of Figure 3, because no explanation about how authors perform the prediction data set and/or prediction method are included. Plus, I can not understand where it they get the extrapolation of soil temperature for the entire work area, as well as, the other variables.
Mc 2.- About modeling procedure. Authors continue using only one modeling method (GAM), when I had recommended the use of other modeling methodologies for contrasting, as EMFA, GLM or MAXENT, all available in R.
. - Minor considerations (mc):
mc 3.- Beetle data. Authors do not improve this section, the new version is identical to the first. So, they wrote that samples were taken from May to September every year (2017-2019), in sampling of 3 days in 124 plots, but I do not understand how often times repeat this sampling was repeated between May and September, once, twice? Monthly, weekly? I think that these data is relevant to understand the quality of the insect data set, composed by 26 species, a scarce number of species considering the big size of work area.
mc 4.- A lack of information of predictors variables: units, maximum and minimum values, scale, etc… continues missing.
mc 5.- Results. Subsection 3.1 Population size is very poorly and imprecise. Authors wrote a mean+- an imprecise number. I do not understand that refer the mean value: number of specimens? Number of species?.
I also recommended the use of more informative biodiversity index as Shannon Index, but authors do no taken into account this recommendation.
I continue thinking that the paper is not suitable for publication in the present stage, and also in my opinion, it could be a lot improved.
Also I continues encouraging to the authors to beside use the useful and free climatic dataset WorldClim (https://www.worldclim.org/), whose could improve their model. Also I encourage to try of include other models as ENFA, GLM or MAXENT.
Author Response
Dear review: please see the attachment
Round 3
Reviewer 1 Report
The manuscript has been improved accordingly recommendations, so, I consider it suitable for publication in Insects